# Sleep Disturbances and Phenoconversion in Patients with REM Sleep Behavior Disorder

**DOI:** 10.3390/jcm10204709

**Published:** 2021-10-14

**Authors:** Hyunjin Jo, Dongyeop Kim, Jooyeon Song, Sujung Choi, Eunyeon Joo

**Affiliations:** 1Department of Neurology, Samsung Medical Center, School of Medicine, Sungkyunkwan University, Seoul 06351, Korea; bell530@naver.com (H.J.); hap2028@naver.com (D.K.); jooyeon.song@samsung.com (J.S.); 2Graduate School of Clinical Nursing Science, Sungkyunkwan University, Seoul 06351, Korea; sujungchoi@hanmail.net

**Keywords:** REM sleep behavior disorder, phenoconversion, sleep, periodic limb movement, obstructive sleep apnea

## Abstract

Objective: We aimed to investigate relationships between sleep disturbances and phenoconversion to neurodegenerative diseases in patients with REM sleep behavior disorder (RBD). Method: Using a comprehensive sleep database in a university-affiliated hospital between December 2014 and March 2021, we reviewed the data of 226 patients with RBD (182 patients with idiopathic RBD (iRBD) and 44 patients with symptomatic RBD (sRBD) with a neurodegenerative disease). Results: Among 226 patients with RBD (male, 61.5%), the mean age at RBD onset and mean disease duration were 59.4 ± 10.5 and 5.9 ± 5.6 years, respectively. Further, 111 (49.1%) patients had periodic limb movements during sleep (PLMS, PLM index ≥ 15/h), while 110 patients (48.7%) had comorbid obstructive sleep apnea (OSA, respiratory disturbance index ≥ 15/h). There was a positive correlation between age at RBD onset and the apnea-hypopnea index and Pittsburgh Sleep Quality Index. Compared to patients with iRBD, patients with sRBD showed a lower N3 sleep (3.3 ± 5.0 vs. 1.6 ± 3.1%, *p* = 0.004) and higher periodic limb movement index (36.3 ± 31.8 vs. 56.9 ± 47.5/h, *p* = 0.021) at the baseline. Among the 186 patients with iRBD, 18 (8.0%) developed neurodegenerative diseases (converters, mean follow-up duration: 2.5 ± 1.6 years) and 164 did not (non-converters, mean follow-up 2.4 ± 2.2 years). There was no significant between-group difference in the demographics and baseline clinical features. Continuous positive airway pressure (CPAP) therapy was prescribed in 101 patients with OSA; among them, 71 (70%) patients agreed to use it. CPAP improved dream enactment behaviors. Conclusion: In our study, 8.0% of patients with iRBD showed phenoconversion within a mean follow-up duration of 2.5 years. Polysomnographic parameters could not predict phenoconversion to neurodegenerative disease. However, approximately half of the patients with RBD presented with significant sleep disorders, including OSA or PLMS. CPAP therapy may alleviate RBD symptoms in patients with RBD-OSA.

## 1. Introduction

Rapid eye movement (REM) sleep behavior disorder (RBD) is a parasomnia characterized by repeated episodes of sleep-related vocalization and/or complex motor behaviors during REM sleep. RBD is often related to REM-related dream content and has been termed dream enactment behaviors (DEBs) [1,2]. Patients with RBD visit sleep clinics mainly because of injuries to themselves or their bed partners resulting from DEBs. These behaviors result from loss of muscle atonia observed during normal REM sleep, which is termed REM sleep without atonia (RSWA) [3].

RBD can be categorized into idiopathic (iRBD) or symptomatic (sRBD) RBD [4]. iRBD often does not involve concomitant neurological conditions while sRBD is related to identifiable underlying etiologies [5], including α-synuclein pathology and other neurodegeneration forms, pontine lesions, and toxic effects of medications.

Both iRBD and sRBD are generally associated with neurodegenerative diseases, especially α-synucleinopathies, including Parkinson’s disease (PD), dementia with Lewy bodies (DLB), and multiple system atrophy (MSA) [2,6,7,8]. Although less common, RBD is also found in non-synucleinopathies such as tauopathy, including progressive supranuclear palsy (PSP) and corticobasal syndrome (CBS) [9].

A large multicenter study on 1280 patients with iRBD reported an annual phenoconversion rate of 6%; moreover, 74% of the patients showed phenoconversion within a 12-year follow-up period. After an average follow-up duration of 4.6 years, 28% of the patients showed phenoconversion; among them, 52%, 43.5%, and 4.5% developed PD, DLB, and MSA, respectively [10]. Therefore, iRBD could represent the prodromal state of these neurodegenerative diseases [11,12].

Sleep disturbance could be an important factor underlying the risk and progression of neurodegenerative diseases such as PD [13,14]. A retrospective cohort study on over 91,000 Taiwanese patients with non-apnea sleep disorders who lacked pre-existing PD found that the presence of sleep disorders was an independent risk factor for incident PD [15]. The mechanistic link between sleep disturbance and neurodegeneration may involve intermittent hypo-oxygenation, inflammation, or protein homeostatic changes [13]. However, sleep-related parameters that may be markers for neurodegenerative disease occurrence remain unclear. We aimed to identify specific sleep disorders or sleep parameters that could contribute to phenoconversion to neurodegenerative disease in patients with RBD. Moreover, we aimed to assess whether treating sleep disorders may change or improve RBD symptoms and disease courses. 

## 2. Methods

### 2.1. Participants

We included patients who visited our hospital for abnormal behavior during sleep and who were diagnosed with RBD as a result of polysomnography (PSG). Those with serious medical conditions (advanced cancer, myocardial infarction, etc.), those with serious neurological or psychological diseases (stroke, myasthenia gravis, amyotrophic lateral sclerosis, etc.), and those who had difficulty in cooperating with the research, such as filling out questionnaires and taking medical history due to cognitive impairment were excluded. In conclusion, we retrospectively enrolled 250 patients diagnosed with RBD based on the clinical history and PSG at a sleep clinic of a university hospital from December 2014 to March 2021. We excluded 7 patients with antidepressant-related RBD, 3 with narcolepsy-associated RBD, and 14 patients with missing detailed records. 

We evaluated the longitudinal course of 182 and 44 patients with iRBD and sRBD, respectively. Patients underwent routine follow-up visits every 3–12 months in our sleep clinic. At every visit, the patients were checked for the occurrence of neurological disorders through detailed history taking and neurological examination by neurologists. In case a patient showed motor or cognitive impairments probably related to neurodegenerative disease, the patient was referred to a movement/dementia clinic to diagnose phenoconversion to defined parkinsonism or dementia. The diagnosis of parkinsonism was made according to Movement Disorder Society (MDS) criteria, and the likeliest underlying diagnosis (i.e., PD or MSA) was determined by a movement disorders specialist. Standardized motor examination was tested with the Korean version of the MDS-Unified Parkinson’s Disease Rating Scale (UPDRS) [16] and/or Unified Multiple System Atrophy Rating Scale (UMSARS) [17]. Additionally, most patients underwent an 18F-FP-CIT PET scan. To evaluate cognitive function, all participants underwent tests with a standardized neuropsychological battery, the Seoul Neuropsychological Screening Battery (SNSB) [18]. Previously accepted criteria were applied for the diagnoses of PD, DLB, MSA, mild cognitive impairment (MCI), and Alzheimer’s disease (AD) [19,20,21,22,23,24].

### 2.2. Clinical Assessment

We obtained information regarding basic demographic factors (age, sex, education level, alcohol, and smoking) through self-report questionnaires. We obtained information regarding RBD-related factors (age at RBD onset, age at RBD diagnosis, RBD duration, DEB presence, DEB frequency, presence of DEB-related injury, presence of hyposmia, medication history, daily doses of RBD medications, concomitant neurodegenerative disease, and age at phenoconversion) through a medical chart review. 

The age at RBD diagnosis was defined as the age when PSG revealed RSWA with or without RBD behaviors. RBD duration was defined as the interval between the reported onset of RBD symptoms and the time of RBD diagnosis. DEB frequency was defined as the number of days in a month that DEB occurred. The severity of injury caused by DEB was defined as a categorical variable ranging from mild (no lasting signs), moderate (bruises), or marked (injury requiring medical attention such as a laceration or fracture) as previously described [25]. The age at phenoconversion was defined as the time of diagnosis based on accepted clinical criteria from experts in movement/dementia clinics.

Olfactory function was examined using the Korean Version of the Sniffin’ Sticks (KVSS) test. Eight different smells were tested, with the patient choosing one smell from four correct answers (0–4, anosmia; 5–6, hyposmia; 7–8, normosmia) [26].

In patients with OSA requiring continuous positive airway pressure (CPAP) therapy, we reviewed data regarding CPAP compliance based on machine download. Adherence can be defined using CPAP use for ≥4 h/night and ≥70% of nights.

### 2.3. Subject Questionnaires

Subjective daytime sleepiness was used to evaluate the Korean version of the Epworth Sleepiness Scale (ESS) [27,28]. An ESS score > 10 was considered as clinically significant daytime sleepiness [27].

The seven-item Korean version of the Insomnia Severity Index (ISI) allows subjective assessment of the severity of insomnia symptoms (difficulty falling asleep, difficulty maintaining sleep, or early morning awakening) and their consequences [29]. The threshold for clinically significant insomnia was set at a cutoff score of 14 [17] (0–7, absence of insomnia; 8–14, sub-threshold insomnia; 15–21, moderate insomnia; 22–28, severe insomnia) [30]. 

Sleep quality and sleep disturbance were assessed using the Korean version of the Pittsburgh Sleep Quality Index (PSQI) [31] during a 1-month period. The total PSQI score has a range of 0–21, with a score > 5 suggesting poor sleep quality and higher scores indicating worse sleep quality [32].

Additionally, depressive symptoms were measured using the Korean version of the Beck Depression Inventory (K-BDI-II), which is a 21-item self-report questionnaire for measuring the severity of depressive symptoms within the previous 2 weeks [33] (0–13, minimal depression; 14–19, mild depression; 20–28, moderate depression; 29–63, severe depression) [34].

### 2.4. Polysomnography

PSG studies were recorded with standard electrodes and sensors using Remlogic (Embla Systems, Denver, CO, USA). We used previously described PSG equipment [35]. We collected the following PSG data: sleep parameters (total sleep time [TST], sleep latency, REM sleep latency, wakefulness after sleep onset, sleep efficiency [SE]), sleep stage (N1, N2, N3, REM sleep, %), apnea-hypopnea index (AHI), respiratory disturbance index (RDI), periodic limb movement index (PLMI), and movement arousal index. The presence of RSWA was defined by the American Academy of Sleep Medicine scoring criteria [36]. Moreover, RBD was diagnosed using the third edition of the International Classification of Sleep Disorders criteria [37]. We calculated the proportion of epochs showing RSWA among epochs of the total REM sleep and used it for analysis (total number of REM epochs with RSWA/total REM sleep epochs × 100) based on the standard rule [36]. DEB observed during REM sleep was video analyzed. As previously described, motor events were classified as minor, major, complex, or scenic behaviors, and vocalizations were classified as mumblings, sounds, or words [38,39].

### 2.5. Statistical Analysis

Statistical analyses were performed using the Statistical Package for Social Sciences (SPSS) for Windows, Version 21.0 (SPSS Inc., Chicago, IL, USA). Statistical significance was set at a two-tailed *p*-value < 0.05. Continuous variables are expressed as mean ± standard deviation. Categorical data are expressed as frequencies and percentages. Data were tested for normality using the Shapiro–Wilk test. Normally and non-normally distributed continuous variables were analyzed using the independent sample *t*-test and Mann–Whitney U-test, respectively. Categorical variables were analyzed using chi-square tests or Fisher’s exact test. As an additional nonparametric test, the Wilcoxon sign rank test was used for paired samples. We performed partial correlation tests to determine the relationship between RBD-related factors and sleep-related parameters. Here, we used the age at PSG and neck circumference as control variables since aging and thick neck circumference can affect sleep fragmentation and indexes, including AHI and PLMI.

## 3. Results

### 3.1. Demographic and Clinical Characteristics in iRBD vs. sRBD 

Demographic and Clinical Characteristics in iRBD vs. sRBD are showed in Table 1.

At the initial assessment, 182 (80.5%) and 44 (19.5%) patients were diagnosed with iRBD and sRBD, respectively.The diagnosed neurological disorders were PD (*n* = 11), MSA (*n* = 11), DLB (*n* = 2), MCI (*n* = 18), and AD *n* = 2). The mean follow-up duration of patients with iRBD and sRBD were 2.0 ± 2.1 and 3.7 ± 3.9 years, respectively. 

There were 162 patients (71.7%) with DEB; moreover, the DEB frequency was 12.3 ± 11.1/m. DEB was observed in various patterns, from benign hand gestures to violent kicking or falling out of the bed. DEB caused injuries to themselves and bed partners in 85 (37.6%) and 39 (17.3%) patients, respectively. The reported severity of self-injury was mild in 18.8%, moderate (bruises or visible injury) in 69.4%, and marked in 11.8%. One patient suffered subdural hematomas after falling out of bed.

Among 100 patients, 58 (58.0%) and 34 (85.0%) patients had subjective hyposmia and objective hyposmia (KVSS I score ≤ 6), respectively. The KVSS I score is negatively correlated with RBD duration (rho = −0.360, *p* = 0.026). However, there were no between-group differences in the KVSS I score and the proportion of patients with objective hyposmia.

Medications for DEB control were prescribed with 120 patients. Among them, 99 (82.5%), 9 (7.5%), 66 (55.0%), 24 (20.0%), and 21 (17.5%) patients controlled DEB with melatonin and/or clonazepam, melatonin only, clonazepam only, melatonin combined with clonazepam therapy, and other drugs, respectively. The mean melatonin and clonazepam doses were 2.0 ± 0.3 and 0.6 ± 0.3 mg, respectively. Medications were given after the initial assessment (PSG, MRI, and questionnaires) for diagnosis.

The mean ESS score of patients with RBD was 7.2 ± 4.4; moreover, 19.0% of the patients had an ESS score > 10. Overall, the mean ISI score of patients with RBD was 9.1 ± 6.3; moreover, 16.4% of them presented with clinically significant insomnia-related symptoms (ISI > 14). The mean PSQI score was 6.6 ± 3.6, with 50.4% of the patients showing higher PSQI scores. However, there was no significant between-group difference in the mean score and proportion of patients with abnormal scores of the ESS, ISI, and PSQI. 

### 3.2. Polysomnographic Findings and RSWA in iRBD vs. sRBD

Polysomnographic findings and RSWA of patients with RBD are showed in Table 2.

PSG data were available for all 226 patients. There were 221 patients with RSWA; among them, 179 patients had DEBs documented by PSG. Among them, 130 patients (72.6%) showed motor events, and the types of motor events were minor in 49 patients (27.4%), major in 82 patients (45.8%), and complex or scenic behaviors in 32 patients (17.9%). In total, 147 patients (82.1%) showed vocalization, and their types were mumbling in 76 patients (42.5%), sounds in 71 patients (39.7%), and words in 50 patients (27.9%). Neither RBD behavior nor RSWA was recorded in five patients; however, one patient showed RBD behaviors with RSWA on follow-up PSG. 

Compared with patients with iRBD, patients with sRBD showed lower N3 sleep (3.3 ± 5.0 vs. 1.6 ± 3.1, *p* = 0.004) and a higher PLMI (36.3 ± 31.8 vs. 56.9 ± 47.5, *p* = 0.021). RSWA was present in 14.0 ± 13.3% of the epochs of REM sleep. RSWA was significantly more present in patients with sRBD than in patients with iRBD (19.1 ± 14.8% vs. 12.8 ± 12.7%, *p* = 0.003) during tonic and phasic phases. 

There were 111 (49.1%) patients with periodic limb movements during sleep (PLMS) with PLMI ≥ 15, including 83 (45.6%) and 28 (63.6%) patients with iRBD and sRBD, respectively.

There were 110 (48.7%) patients with comorbid OSA (defined as an RDI ≥ 15/h). There was no significant between-group difference in the presence of OSA (47.3% vs. 54.5%). CPAP was prescribed in 101 (44.7%) patients (mean AHI 27.6 ± 17.6/h); among them, 71 accepted and 30 refused, respectively, to use CPAP (acceptance rate 70.3%, CPAP use duration 18.2 ± 20.4 months). Among the 71 patients who used CPAP, 40 (56.3%) and 31 (43.7%) were adherent (mean CPAP duration: 22.3 ± 23.2 months) and non-adherent (mean CPAP duration: 13.5 ± 15.4 months) to CPAP use, respectively. The DEB reduction rate was 35.0%, 27.0%, and 13.3% in the CPAP adherent, CPAP non-adherent, and no-CPAP groups, respectively. After excluding patients who underwent concomitant drug treatment with CPAP (to exclude the effect of drug therapy), the DEB reduction rate was 26.9% and 21.1% in the CPAP adherent and CPAP non-adherent groups, respectively; however, no patients in the no-CPAP group showed reduced DEB frequency (Figure 1). Compared with the no-CPAP group, the CPAP group (regardless of adherent and non-adherent) showed improvement in DEB.

After adjusting for age at the PSG study and neck circumference, there was a positive partial correlation of age at RBD onset with the PSQI score, AHI, and RDI (rho = 0.810, *p* = 0.015; rho = 0.729, *p* = 0.040; and rho = 0.795, *p* = 0.018, respectively) (Appendix A Appendix A).

### 3.3. Converters vs. Non-Converters 

Comparison between converters and non-converters in patients with iRBD is showed in Table 3.

During the follow-up period, six patients with iRBD showed phenoconversion to PD (*n* = 1), MSA (*n* = 1), MCI (*n* = 2), and AD (*n* = 2). Additionally, 12 patients developed motor or cognitive symptoms suspected of phenoconversion (prior to accurate diagnosis from a movement or dementia clinic). Moreover, five patients with MCI showed phenoconversion to PD (*n* = 2), DLB (*n* = 2), and AD (*n* = 1) during the follow-up period. Finally, during the last follow-up visit, there were 164 patients with iRBD, 50 patients with sRBD (PD (*n* = 14), MSA (*n* = 12), DLB (*n* = 4), MCI (*n* = 15), and AD (*n* = 5), and 12 patients had suspected phenoconversion (Figure 2). 

There were 18 (8.0%) patients with iRBD who converted to sRBD after a mean interval of 3.7 ± 2.4 years from the onset of RBD symptoms (mean follow-up period: 2.5 ± 1.6 years) while 164 patients did not show phenoconversion at the last follow-up visit (mean 2.4 ± 2.2 years).

There was no significant difference in the demographics or clinical characteristics, including hyposmia, between converters and non-converters at the baseline. There was a higher proportion of patients with clinically significant insomnia among converters than among non-converters (22.2% vs. 14.6%, *p* = 0.006). However, there were no between-group differences in the ESS, PSQI, and K-BDI-II scores, as well as in the RSWA. There were no between-group differences in the PSG (at the time of diagnosis) measures except for the REM arousal index. Non-converters showed higher REM arousal index than converters (17.1 ± 10.0 vs. 11.2 ± 4.9, *p* = 0.031). 

Among the 164 non-converters, 74 (45.1%) and 76 (46.3%) patients had PLMS and OSA, respectively. Among the 18 converters, 9 (50.0%) and 10 (55.6%) had PLMS and OSA, respectively. However, there was no significant between-group difference in the concomitant ratio of PLMS and OSA. Among the non-converters, PSG was repeated in 19 patients at a mean interval of 2.8 ± 2.1 years, with no between-group differences in their baseline and follow-up PSG parameters (Appendix A Appendix A). 

## 4. Discussion

This longitudinal study examined the sleep-related features, including PSG findings, in patients with RBD. Although we could not identify sleep-related parameters that could predict phenoconversion in patients with iRBD, there was an association between neurodegenerative disease and sleep disturbances such as decreased slow wave sleep (SWS) and high PLMI. Additionally, comorbid sleep disorders, including PLMS and OSA, are common in patients with RBD. Specifically, CPAP, which is a treatment option for OSA, can allow the alleviation of RBD symptoms.

### 4.1. Subjective Sleep Complaints (Self-Reported Questionnaires)

In our cohort, there was a considerable proportion of patients with RBD with insomnia and poor sleep quality (high ISI and PSQI scores). There was a significantly higher proportion of patients with insomnia among converters than among non-converters; however, our small sample size cannot yield conclusive findings. Baseline daytime sleepiness (ESS) or subjective sleep quality scores (PSQI) did not predict phenoconversion in patients with iRBD.

Daytime sleepiness and insomnia are common characteristics of neurodegenerative synucleinopathies. Somnolence occurs in 30–40% and 40–70% of patients with PD and DLB, respectively, and becomes more common with disease progression [40]. Insomnia occurs in up to 50% of patients with PD and is a common early disease characteristic [41]. The glymphatic system function that clears the accumulation of abnormal proteins increases during sleep time rather than wake time. Therefore, insufficient sleep may interfere with the clearance of abnormally accumulated protein in the brain, thus promoting the development of neurodegenerative disease [42,43]. However, there is limited evidence for somnolence and insomnia as prodromal markers. A recent study on iRBD cohorts confirmed that somnolence and insomnia could not predict neurodegeneration [44]. Our findings are consistent with the hypothesis that subjective sleep quality evaluation using ESS, ISI, and PSQI scores could not predict neurodegeneration.

### 4.2. Sleep Architecture

Sleep continuity and architecture are disturbed in patients with iRBD. Meta-analyses revealed significantly reduced TST and SE, as well as increased sleep latency and SWS, in patients with iRBD [45]. Additionally, patients with PD showed frequent awakenings, low SE, decreased SWS and REM sleep, and increased light sleep and REM latency [46,47]. In our study, patients with sRBD showed a greater decrease in SWS than patients with iRBD; however, there was no significant between-group difference in age at the time of PSG. Moreover, there was no difference between converters and non-converters. Therefore, although there is an association of decreased SWS with neurodegeneration, it cannot predict phenoconversion. 

These sleep disturbances could be attributed to neuropathological changes. The dopamine system is crucial for regulating sleep and waking [48,49]. Moreover, since the brainstem, which is initially involved in synucleinopathies, is crucially involved in sleep and circadian behavior, sleep disturbances in RBD could be considered as neurodegenerative disease symptoms. 

### 4.3. Comorbid Sleep Disorders

#### 4.3.1. Periodic Limb Movements during Sleep

Periodic limb movements are common and occurred in 49.1% of our patients with RBD (PLMI ≥ 15). This is consistent with previous reports of high PLMS comorbidity in patients with iRBD. Several large case series on patients with RBD reported the presence of PLMS (with PLMI > 10–20) in approximately 47~80% of patients with RBD [50,51,52,53].

The high PLMS occurrence in patients with RBD suggests that PLMS and RBD have a partly common pathogenesis. The absence of brainstem inhibition on spinal motoneurons, which is the main characteristic of RBD, could be attributed to the presence of PLMS [54]. Additionally, PLMS could be attributed to impaired central dopaminergic transmission [52]. 

Notably, Schenck et al. [55] reported a higher PLMI in patients with iRBD who developed parkinsonism than in patients who remained idiopathic after 6 years (85.2 ± 44.8 vs. 35.9 ± 21.1, *p* = 0.003). Therefore, in patients with iRBD, increased PLMI could be a predictor of PD [56]. 

Patients with sRBD had a higher PLMI than patients with iRBD (56.9 ± 47.5 vs. 36.3 ± 31.8, *p* = 0.021); however, there was no difference between converters and non-converters, which could be attributed to the RBD duration. PLMS (especially during stage REM) might reflect the length of RBD morbidity, which could be related to the RBD disease process [57]. Schenck et al. conducted PSG about 9 years after RBD onset; contrastingly, in our study, PSG was performed about 5 years after RBD onset. Specifically, converters had a short RBD duration of 3.9 years. The RBD duration required for differences in PLMI between converters and non-converters remains unclear. To confirm this, there is a need for a longitudinal PSG follow-up study on patients with iRBD.

#### 4.3.2. Obstructive Sleep Apnea

OSA is a common disorder characterized by repetitive episodes of upper airway obstruction while sleeping, which can cause hypoxemia and sleep fragmentation [58]. OSA and RBD have different pathophysiological mechanisms and clinical manifestations. However, DEBs can occur among patients with OSA during arousals from NREM and REM sleep, which is clinically termed pseudo-RBD [1]. Additionally, clonazepam, which is the treatment of choice for RBD, may worsen OSA. Therefore, it is important to identify concomitant OSA in RBD.

Few studies have discussed comorbid OSA in patients with RBD. Previous studies have reported an OSA prevalence of 34.0% (AHI > 10/h) [50], 41.0% (AHI > 15) [59], and 89.2% (AHI ≥ 5/h) [60], which vary depending on the diagnostic criteria for OSA. Since OSA occurred in 48.7% (RDI ≥ 15/h) of our patients, the coexistence of RBD and OSA is not uncommon.

Two studies have discussed the therapeutic effect of CPAP on concomitant OSA. In one study, 8 (62%) out of 13 patients with OSA underwent CPAP treatment; among them, 6 (75%) reported improvement in RBD symptoms after CPAP use [59]. In another study, nearly half (45.8%) of 27 CPAP users reported improved RBD symptoms regardless of CPAP compliance [60].

Theoretically, OSA could cause both pseudo and true RBD symptoms; moreover, OSA could exacerbate RBD through sleep fragmentation, arousals, and sleep disruption [61]. Therefore, the improvement of RBD symptoms after CPAP use could be attributed to reduced pseudo-RBD behaviors or decreased sleep fragmentation [58,60].

In our study, CPAP users reported a decreased frequency of DEB, regardless of CPAP compliance. Although our findings were not statistically significant, since our findings were consistent with previous reports, CPAP treatment in RBD could help alleviate RBD symptoms. 

## 5. Limitations and Strengths

This study has several limitations. First, since this is a retrospective study by chart review, some patients had missing information. Additionally, we used reports of patients and their family members, which could have introduced recall bias. Recall bias may apply to numerous RBD-related factors, including disease onset, disease duration, and DEB frequency. Second, the relatively short follow-up duration could not allow the determination of the long-term phenoconversion rate of patients with iRBD. There is a need for future studies with a long-term follow-up period. Third, since this was a single-center study, the generalizability of our results is limited. However, given our large sample size and the consistency of our results with previous reports, these limitations can be offset to some extent. Fourth, because of the night-to-night variability of RBD symptoms and the first night effect, a one-night PSG may not be sufficient for analysis. Finally, there were limitations in determining the phenotype of neurodegenerative disease in converters. During the primary years of neurodegenerative disease, it is difficult to determine the phenotype due to a lack of efficient methods differentiating PD (synucleinopathy) and PSP-P (tauopathy) [62].

Regarding the strengths of our study, we analyzed sleep-related characteristics in patients with RBD and determined whether treating accompanying sleep disorders could reduce RBD symptoms. Our findings highlight the importance of diagnosis and treatment of comorbid sleep disorders in patients with RBD. Additionally, our study had a relatively large sample size. Specifically, we included a larger number of CPAP users with RBD and OSA than those in previous studies. Although our findings were not statistically significant, they can inform suggestions of additional treatment options for RBD. 

## 6. Conclusions

In our cohort, approximately 8.0% of patients with iRBD showed phenoconversion after approximately 2.5 years of follow-up. Although there was no difference in sleep-related characteristics between non-converters and converters, the accompanying neurodegenerative disease was associated with reduced SWS and higher PLMI. Additionally, concomitant PLMS and OSA were common in patients with RBD. Specifically, since RBD develops at a later age, there is increased comorbidity of OSA, which may worsen sleep disturbance. Although not statistically significant, CPAP for treating OSA could reduce RBD symptoms. Therefore, there is a need to identify and treat comorbid OSA in patients with RBD to improve their quality of life. 

## Figures and Tables

**Figure 1 jcm-10-04709-f001:**
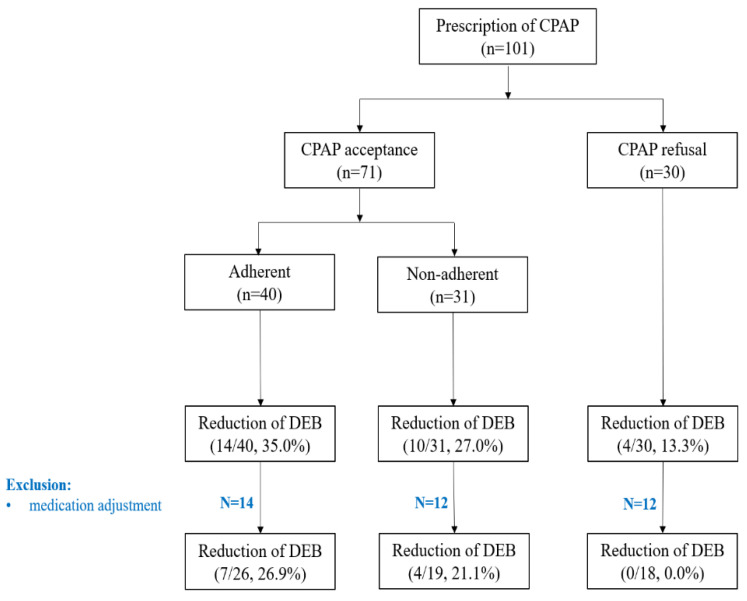
Flow chart of CPAP use.

**Figure 2 jcm-10-04709-f002:**
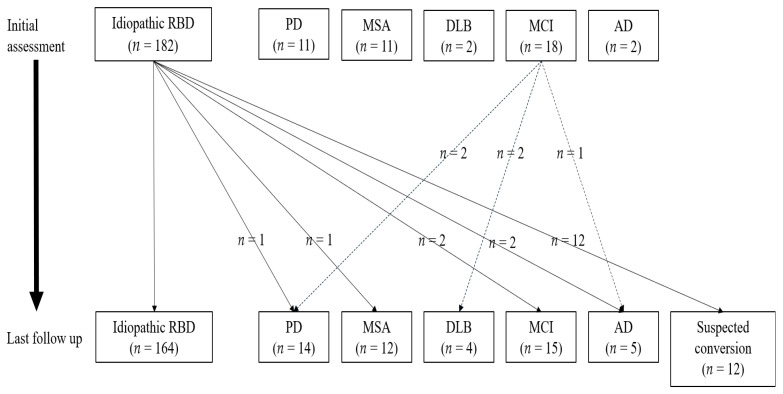
Flow chart describing patients with idiopathic and symptomatic REM sleep behavior disorder.

**Table 1 jcm-10-04709-t001:** Demographic and clinical characteristics of patients.

Variables	Total (*n* = 226)	IRBD (*n* = 182)	SRBD (*n* = 44)	*p*-Value (IRBD vs. SRBD)
Male, *n* (%)	139 (61.5)	110 (60.4)	29 (65.9)	0.605 ^†^
Education, *n* (%)				
≤12 years	52 (23.0)	40 (22.0)	12 (27.3)	0.428 ^†^
≥13 years	159 (70.4)	130 (71.4)	29 (65.9)	
Alcohol, *n* (%)	97 (42.9)	81 (46.8)	16 (38.1)	0.388 ^†^
Smoking, *n* (%)	20 (8.8)	17 (9.7)	3 (7.1)	0.771 ^†^
Age at RBD onset, years	59.4 ± 10.5	59.4 ± 11.1	59.0 ± 7.9	0.834
Age at RBD diagnosis, years	65.5 ± 9.9	65.0 ± 10.3	67.6 ± 7.7	0.107
RBD duration, years	5.9 ± 5.6	5.5 ± 5.6	7.5 ± 5.4	0.050
Follow-up duration, years	2.3 ± 2.6	2.0 ± 2.1	3.7 ± 3.9	0.010 *
Presence of DEB, *n* (%)	162 (71.7)	136 (74.7)	26 (59.1)	0.417 ^†^
Frequency of DEB, days/month				
before treatment	12.3 ± 11.1	11.5 ± 10.6	16.5 ± 12.8	0.130
after treatment	4.1 ± 5.3	3.6 ± 3.6	6.0 ± 9.3	0.435
DEB related patient injury, *n* (%)	85 (37.6)	72 (39.6)	13 (29.5)	0.279 ^†^
DEB related bed-partner injury, *n* (%)	39 (17.3)	35 (19.2)	4 (9.1)	0.402 ^†^
Injury type, n, (% of injured patients)				
Mild	16 (18.8)			
Moderate	59 (69.4)			
Marked	10 (11.8)			
KVSS I score	4.9 ± 1.7	4.8 ± 1.7	5.6 ± 0.9	0.337
KVSS I ≤ 6, *n* (%)	34 (85.0%)	30 (85.7%)	4 (80.0%)	1.000 ^†^
Medical treatment				
Melatonin, number of patients	32	22	10	
dose	2.0 ± 0.3	1.9 ± 0.4	1.9 ± 0.4	0.654
Clonazepam, number of patients	92	77	15	
dose	0.6 ± 0.3	0.6 ± 0.3	0.7 ± 0.2	0.232
Family history, *n* (%)				
RBD	8 (3.5)	5 (2.7)	3 (6.8)	
Dementia	5 (2.0)	5 (2.7)	0	
PD	5 (2.0)	4 (2.2)	1 (2.3)	
BMI, kg/m^2^	24.4 ± 3.1	24.4 ± 3.1	24.3 ± 3.3	0.936
NC, cm	37.1 ± 4.0	36.9 ± 3.4	36.1 ± 3.0	0.213
ESS	7.2 ± 4.4	7.3 ± 4.5	7.1 ± 4.0	0.766
ESS > 10, *n* (%)	43 (19.0%)	34 (18.7)	9 (20.5)	0.773 ^†^
ISI	9.1 ± 6.3	9.0 ± 6.2	9.7 ± 6.8	0.491
ISI > 14, *n* (%)	37 (16.4%)	28 (15.4	9 (20.5%)	0.415 ^†^
PSQI	6.6 ± 3.6	6.6 ± 3.5	6.8 ± 4.0	0.713
PSQI > 5, *n* (%)	114 (50.4%)	96 (52.7)	18 (40.9)	0.259 ^†^
K-BDI-II	13.7 ± 9.5	13.4 ± 9.4	15.0 ± 9.8	0.326

RBD, REM sleep behavior disorder (RBD); DEB, dream enactment behavior; KVSS, Korean Version of Sniffin’ Sticks; PD, Parkinson’s disease; ESS, Epworth sleepiness scale; ISI, insomnia severity index; PSQI, Pittsburgh sleep quality index; BDI-II, Beck depression inventory-II; * *p*-value < 0.05. ^†^ Variables were analyzed by Fisher’s exact test. Other comparisons were tested by an independent sample *t*-test.

**Table 2 jcm-10-04709-t002:** Polysomnographic findings and RSWA of patients with RBD.

Variables	Total (*n* = 226)	IRBD (*n* = 182)	SRBD (*n* = 44)	*p*-Value (IRBD vs. SRBD)
Total sleep time, min	356.2 ± 65.4	359.6 ± 63.2	342.2 ± 72.6	0.111
Sleep latency, min	20.3 ± 30.0	20.8 ± 32.0	18.2 ± 19.2	0.606
REM latency, min	108.6 ± 64.4	104.1 ± 59.9	127.1 ± 78.7	0.075
WASO, %	16.5 ± 11.4	15.7 ± 10.3	19.7 ± 14.9	0.138
Sleep efficiency, %	79.6 ± 12.8	80.3 ± 11.9	76.9 ± 16.0	0.186
Sleep stages				
N1 sleep, %	20.4 ± 10.6	20.0 ± 10.5	22.3 ± 11.0	0.197
N2 sleep, %	55.1 ± 11.1	54.9 ± 11.2	56.1 ± 10.7	0.490
N3 sleep, %	3.0 ± 4.7	3.3 ± 5.0	1.6 ± 3.1	0.004 *
REM sleep, %	21.5 ± 6.9	21.9 ± 6.5	20.0 ± 8.0	0.097
Arousal Index, /h	20.0 ± 9.3	20.1 ± 9.4	19.9 ± 9.1	0.897
REM arousal index, /h	16.1 ± 9.8	16.1 ± 10.0	16.2 ± 9.0	0.938
AHI, /h	16.5 ± 16.3	15.3 ± 14.4	21.7 ± 21.9	0.071
REM AHI, /h	16.9 ± 18.2	15.7 ± 17.1	22.0 ± 21.8	0.063
RDI, /h	18.8 ± 16.1	17.7 ± 14.4	23.5 ± 21.4	0.090
RDI ≥ 15, *n* (%)	110 (48.7)	86 (47.3)	24 (54.5)	0.405 ^†^
REM RDI, /h	18.8 ± 18.0	17.7 ± 17.0	23.7 ± 21.2	0.069
PLMI	40.8 ± 36.7	36.3 ± 31.8	56.9 ± 47.5	0.021 *
PLMI ≥ 15, *n* (%)	111 (49.1)	83 (45.6)	28 (63.6)	0.136 ^†^
MAI	2.1 ± 3.7	2.1 ± 3.9	1.9 ± 2.4	0.729
RSWA (%)	14.0 ± 13.3	12.8 ± 12.7	19.1 ± 14.8	0.003 *
Tonic activity	3.2 ± 6.5	3.1 ± 7.0	3.9 ± 5.4	0.019 *
Phasic activity	10.5 ± 9.8	9.7 ± 8.8	15.2 ± 13.2	0.005 *

RBD, REM sleep behavior disorder (RBD); BMI, body mass index; NC, neck circumference; WASO, wake after sleep onset; REM, rapid eye movement; NREM, non-rapid eye movement; AHI, apnea-hypopnea index; RDI, respiratory disturbance index; PLMI, periodic limb movements of sleep index; MAI, movement arousal index; RSWA, REM sleep without atonia * *p*-value < 0.05. ^†^ Variables were analyzed by Fisher’s exact test. Other comparisons were tested by an independent sample *t*-test.

**Table 3 jcm-10-04709-t003:** Comparison between converters and non-converters in patients with iRBD.

	Non-Converters (*n* = 164)	Converters (*n* = 18)	*p*-Value
Clinical characteristics			
Male, *n* (%)	101 (63.1)	10 (55.6)	0.610 ^†^
Age at RBD onset, years	59.6 ± 10.7	61.3 ± 8.0	0.848
Age at RBD diagnosis, years	65.3 ± 9.4	62.3 ± 12.8	0.410
RBD duration, years	5.7 ± 5.8	3.7 ± 2.4	0.433
Follow-up duration, years	2.4 ± 2.2	2.5 ± 1.6	0.544
KVSS I score	5.2 ± 1.5	5.5 ± 2.1	0.821
KVSS I ≤ 6, *n* (%)	26 (66.7%)	4 (80.0%)	0.342 ^†^
Medical treatment			
Melatonin, number of patients	18	4	
dose	1.9 ± 0.5	2.0 ± 0.0	0.758
Clonazepam, number of patients	69	8	
dose	0.6 ± 0.3	0.7 ± 0.4	0.381
Self-reported sleep measures			
ESS	8.0 ± 4.9	6.7 ± 5.2	0.134
ESS > 10, *n* (%)	43 (26.2)	3 (16.7)	0.757 ^†^
ISI	9.0 ± 6.1	12.1 ± 6.9	0.066
ISI > 14, *n* (%)	24 (14.6)	4 (22.2)	0.006 ^†^
PSQI	6.5 ± 3.424 (14.6)	7.6 ± 3.4	0.100
PSQI > 5, *n* (%)	84 (51.2)	12 (66.7)	0.341 ^†^
K-BDI-II	13.8 ± 9.3	15.2 ± 8.6	0.412
Anthropometric and Polysomnographic data			
BMI, kg/m^2^	24.1 ± 3.1	25.5 ± 3.5	0.412
NC, cm	36.9 ± 4.1	36.6 ± 2.8	0.469
Total sleep time, min	360.5 ± 65.1	351.5 ± 66.5	0.527
Sleep latency, min	16.8 ± 21.9	16.3 ± 17.6	0.804
REM latency, min	109.9 ± 68.1	91.6 ± 46.7	0.581
WASO, %	16.9 ± 12.3	14.2 ± 8.1	0.651
Sleep efficiency, %	79.8 ± 12.9	80.8 ± 11.6	0.830
Sleep stages			
N1 sleep, %	22.0 ± 11.5	19.8 ± 7.4	0.612
N2 sleep, %	54.0 ± 11.5	54.8 ± 12.0	0.657
N3 sleep, %	2.7 ± 4.1	2.5 ± 6.2	0.914
REM sleep, %	21.3 ± 6.9	23.0 ± 8.7	0.499
Arousal Index, /h	21.1 ± 9.9	20.2 ± 8.3	0.734
REM arousal index, /h	17.1 ± 10.0	11.2 ± 4.9	0.031 *
AHI, /h	17.3 ± 17.0	15.0 ± 12.1	0.836
REM AHI, /h	17.7 ± 18.6	16.1 ± 15.6	0.886
RDI, /h	19.5 ± 16.6	17.9 ± 11.6	0.885
RDI ≥ 15, *n* (%)	76 (46.3)	10 (55.6)	0.352 ^†^
REM RDI, /h	19.6 ± 18.5	17.6 ± 15.4	0.823
PLMI	40.1 ± 37.0	39.9 ± 28.6	0.803
PLMI ≥ 15, *n* (%)	74 (45.1)	9 (50.0)	1.000 ^†^
MAI	2.1 ± 3.9	1.6 ± 2.5	0.716
RSWA (%)	12.4 ± 12.4	16.1 ± 14.9	0.299
Tonic activity	2.8 ± 6.5	5.8 ± 10.4	0.121
Phasic activity	9.6 ± 8.9	10.3 ± 8.8	0.683

RBD, REM sleep behavior disorder (RBD); KVSS, Korean Version of Sniffin’ Sticks; ESS, Epworth sleepiness scale; ISI, insomnia severity index; PSQI, Pittsburgh sleep quality index; BDI-II, Beck depression inventory-II; BMI, body mass index; NC, neck circumference; WASO, wake after sleep onset; REM, rapid eye movement; NREM, non-rapid eye movement; AHI, apnea-hypopnea index; RDI, respiratory disturbance index; PLMI, periodic limb movements of sleep index; MAI, movement arousal index; RSWA, REM sleep without atonia * *p*-value < 0.05. ^†^ Variables were analyzed by Fisher’s exact test. Other comparisons were tested by the Mann–Whitney U-test.

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
