# Peer review of "Sleep Disturbances and Phenoconversion in Patients with REM Sleep Behavior Disorder"

_jcm, 2021, doi:10.3390/jcm10204709_

Round 1

Reviewer 1 Report

Authors elaborate on REM behavior disorder and possible associations between features correlated with this issue.

Issues which could be improved are as follows:

  • in the introduction authors state: "The mechanistic
    link between sleep disturbance and neurodegeneration may involve intermittent hypo

    oxygenation, inflammation, or protein homeostatic changes.
    "-
    this issue should be more widely described
  • the elaboration regarding RBD and its correlation with neurodegenerative disorders should mention the presence of RBD in tauopathic parkinsonisms. Though RBD is more common in synucleinopathies, it may also have an impact on the course of PSP and CBS [1]. In this context due to the lack of efficient methods differentiating PD and PSP-P in the primary years of PD, authors should include this issue in the limitation section [2].
  1. Abbott SM, Videnovic A. Sleep Disorders in Atypical Parkinsonism. Mov Disord Clin Pract. 2014 Jun 1;1(2):89-96. doi: 10.1002/mdc3.12025. PMID: 24955381; PMCID: PMC4061748.
  2. Alster P, Madetko N, Koziorowski D, Friedman A. Progressive Supranuclear Palsy-Parkinsonism Predominant (PSP-P)-A Clinical Challenge at the Boundaries of PSP and Parkinson's Disease (PD). Front Neurol. 2020 Mar 10;11:180. doi: 10.3389/fneur.2020.00180. PMID: 32218768; PMCID: PMC7078665.
  • authors should reference the more recent DLB criteria from 2017 [3]

McKeith IG, Boeve BF, Dickson DW, Halliday G, Taylor JP, Weintraub D, Aarsland D, Galvin J, Attems J, Ballard CG, Bayston A, Beach TG, Blanc F, Bohnen N, Bonanni L, Bras J, Brundin P, Burn D, Chen-Plotkin A, Duda JE, El-Agnaf O, Feldman H, Ferman TJ, Ffytche D, Fujishiro H, Galasko D, Goldman JG, Gomperts SN, Graff-Radford NR, Honig LS, Iranzo A, Kantarci K, Kaufer D, Kukull W, Lee VMY, Leverenz JB, Lewis S, Lippa C, Lunde A, Masellis M, Masliah E, McLean P, Mollenhauer B, Montine TJ, Moreno E, Mori E, Murray M, O'Brien JT, Orimo S, Postuma RB, Ramaswamy S, Ross OA, Salmon DP, Singleton A, Taylor A, Thomas A, Tiraboschi P, Toledo JB, Trojanowski JQ, Tsuang D, Walker Z, Yamada M, Kosaka K. Diagnosis and management of dementia with Lewy bodies: Fourth consensus report of the DLB Consortium. Neurology. 2017 Jul 4;89(1):88-100. doi: 10.1212/WNL.0000000000004058. Epub 2017 Jun 7. PMID: 28592453; PMCID: PMC5496518

  • I can't see the recent criteria of diagnosis of PD in the reference section of the manuscript - in my opinion reference nr 15 is not adequate
  • are "7 patients with antidepressant-related RBD, 3 with narcolepsy-95 associated RBD, and 14 patients with missing detailed records." the only exclusion criteria?
  • "In case a patient showed motor 100
    or cognitive impairments probably related to neurodegenerative disease, the patient was 101

    referred to a movement/dementia clinic for detailed evaluation
    " - information concerning how the detailed evaluation was performed should be added

Reviewer 2 Report

The authors aim was to identify specific sleep disorders or sleep parameters that could contribute to phenoconversion to neurodegenerative disease in patients with RBD.

Line 107: inclusion and exclusion criteria are missing.

Line 109: DEB is mentioned for the first time and should be spelled out. DEB should be explained in detail: was only DEB scored when a dream was reported? How was this confirmed and differentiated from other parasomnia or epileptic behavior.

Line 114: How did authors decide on onset of RBD if any kind of nocturnal behavior was preceeding RBD or DEB.

Table 1: please write out: DEB days per month. This value seems to be extremely high for a population reporting DEB. Please relate to other data in discusssion. Please list kind of injury patients reported and try to categorize in order to be able to evaluate the severity of DEB.

How was alcohol consumption defined?

Line 186: was vocalization considered as a sign of RBD? Slight hand movement have been described by the Insbruck group in many healthy persons in REM sleep. It would be good to know how many patients reported the different behaviors.

Line 193: in terms of evaluating the progression it would be good to know how many patients in which group had what dose of medication and at which time it was assessed after the diagnosis.

Table 2: BMI belongs to demographics. N3% is extremely low even for this age group. Please comment.

Did you perform 1 or 2 nights of PSG?

Line 228:….among them 71 accepted and 30 refused…..

Line 232: how much was the reduction rate in each group? Would be interesting to know the influence of CPAP treatment on RBD

Line 289: decrease of N3% cannot be called comorbidity. Please change. PLMI should be PLMD if correct.

Chapter 4.1: questionnaire data should be underpinned by PSG data. Relate to literature on insomnia and accululation of alpha-synuclein, tau protein etc.

Chapter 4.2: There is no control group. Therefore it is difficult to estimate the meaning of PSG outcomes, espcially if only 1 PSG was performed (first night effect)

Line 363: I recommend to discuss this result together with the ISS results

Round 2

Reviewer 1 Report

I have no further comments.

Reviewer 2 Report

The reviewers comments have been answered sufficiently